# Quantification of Empty, Partially Filled and Full Adeno-Associated Virus Vectors Using Mass Photometry

**DOI:** 10.3390/ijms241311033

**Published:** 2023-07-03

**Authors:** Christina Wagner, Felix F. Fuchsberger, Bernd Innthaler, Martin Lemmerer, Ruth Birner-Gruenberger

**Affiliations:** 1Analytical Development Europe, Takeda Vienna, 1220 Vienna, Austria; christina.wagner@takeda.com (C.W.);; 2Gene Therapy Process Development Europe, Takeda Orth an der Donau, 2304 Orth an der Donau, Austria; 3Institute of Chemical Technologies and Analytics, Technische Universität Wien, 1040 Vienna, Austria

**Keywords:** adeno-associated virus vectors, partially filled particles, single-molecule mass photometry, genomic cargo, analytical ultracentrifugation

## Abstract

Adeno-associated viruses (AAV) are one of the most commonly used vehicles in gene therapies for the treatment of rare diseases. During the AAV manufacturing process, particles with little or no genetic material are co-produced alongside the desired AAV capsid containing the transgene of interest. Because of the potential adverse health effects of these byproducts, they are considered impurities and need to be monitored carefully. To date, analytical ultracentrifugation (AUC), transmission electron microscopy (TEM) and charge-detection mass spectrometry (CDMS) are used to quantify these subspecies. However, they are associated with long turnaround times, low sample throughput and complex data analysis. Mass photometry (MP) is a fast and label-free orthogonal technique which is applicable to multiple serotypes without the adaption of method parameters. Furthermore, it can be operated with capsid titers as low as 8 × 10^10^ cp mL^−1^ with a CV < 5% using just 10 µL total sample volume. Here we demonstrate that mass photometry can be used as an orthogonal method to AUC to accurately quantify the proportions of empty, partially filled, full and overfull particles in AAV samples, especially in cases where ion-exchange chromatography yields no separation of the populations. In addition, it can be used to confirm the molar mass of the packaged genomic material in filled AAV particles.

## 1. Introduction

Recombinant adeno-associated virus (rAAV) vectors are one of the leading transport vehicles for the introduction of foreign genetic material in vivo to treat severe and rare diseases in patients. The reason for this lies in the characteristics of rAAV vectors, which stand out by their non-pathogenicity, low immunogenicity and long-term gene expression [1,2,3]. The vast variety of serotypes allows the targeting of specific types of tissues and cells. Furthermore, genetically engineered variants of rAAV vectors to overcome immunological barriers become more attractive and expand the gene therapy portfolio [4]. The high potential of rAAVs in human gene therapy is mirrored in the great number of clinical trials which are currently being investigated worldwide [5]. In addition, three AAV-based and one lentivirus-based gene therapeutics have been approved by the US Food and Drug Administration (FDA) or the European Medicines Agency (EMA) and are commercially available [6,7,8].

AAVs are members of the *Parvoviridae* family and belong to the genus *Dependoparvovirus* [4]. For efficient replication, they require a helper virus, such as Adenovirus (AV) or Herpes Simplex Virus (HSV) [9,10]. The non-enveloped, icosahedral AAV capsid is made up of 60 protein monomers, which can be grouped into three viral protein isoforms: VP1, VP2 and VP3, occurring in a proportion of 5:5:50 [11]. VP1, VP2 and VP3 share a common region, which is defined as VP3. The latter represents the shortest protein sequence variant with the lowest molecular weight of 62 kDa. VP1, the longest protein sequence variant, comprises VP2 and differs from VP2 by an N-terminal extension. VP1 and VP2 have molar masses of 87 kDa and 73 kDa, respectively [12,13,14]. The cargo capacity of the virus is limited to a single-stranded DNA of ~4.7 kb, which is less than for other gene delivery vehicles, such as AV or HSV, which encapsulate double-stranded DNA up to ~8 kb and ~40 kb, respectively [15,16]. To date, there are 13 different AAV serotypes varying in their cellular tropisms due to differences in the capsid surface as a consequence of capsid assembling [17].

During the manufacturing process of AAVs, numerous capsids either lacking genetic cargo (empty), containing truncated versions of the transgene (partially filled) or encapsulating genetic material beyond 4.7 kb (overfill) are co-produced next to the desired AAV vectors loaded with the intact genome of interest (full). Empty, partially filled and overfilled capsids in the final drug product pose a potential risk to the patient and are considered impurities [18,19]. Therefore, it is of high priority to carefully monitor and remove these unwanted byproducts. While the ratio of empty-to-full capsids can be determined easily using, e.g., ion-exchange chromatography in combination with UV detection at 260 and 280 nm, the proportions of partially filled or overfilled particles remain unknown due to the limited resolution of the chromatography system. To date, these subspecies are assessed using analytical ultracentrifugation (AUC), charge-detection mass spectrometry (CDMS) or transmission electron microscopy (TEM). Despite being frequently used in the biopharmaceutical industry, these analytical platforms lack low turnaround times and high sample throughput [20,21,22]. In addition, AUC demands large quantities of purified samples of ~500 µL and genome titers up to 5 × 10^12^ vg mL^−1^ [23]. Compared to AUC, CDMS is less time-intensive (2 h instead of 6 h), but the technique is still in development [21]. Furthermore, CDMS requires specialized equipment and well-trained personnel [23].

Mass photometry provides a faster and more straightforward analysis of AAV subspecies by single-molecule counting. It allows the determination of the full-empty ratio (F/E), the molecular weight of empty and filled AAV capsids as well as the mass of the encapsulated genetic material. Moreover, it can resolve and quantify partially filled and overfilled particles. It does not require any laborious sample preparation, including labelling or immobilization of the analyte [24,25]. Equipped with a robotic arm, the MP can be operated as an automated instrument, facilitating analysis of large sample numbers, which provides MP with a tremendous advantage over orthogonal analytical methods, such as AUC, CDMS or TEM. The principle is based on the interference of light scattered by particles reversibly attaching to a microscope glass slide and light reflected by the cover glass. At the interface between the solid cover slide and the sample liquid, a difference in the contrast between empty and filled particles is observed when binding to the glass surface [26]. The intensity of scattered light is directly proportional to the molar mass of the particle [27].

Here we present a simple alternative analytical technique to AUC with no sample preparation for the quantification of empty, partially filled, filled and overfull AAV capsids using minimal sample volumes of 10 µL and capsid titers around 1.0 × 10^11^ cp mL^–1^. Furthermore, the single molecule events-based approach allows the determination of the molecular weight of the genomic cargo of an AAV product at different stages of AAV manufacturing. It provides excellent comparability with AUC data and good linearity of capsid titers between 4 × 10^10^ cp mL^−1^ and 8 × 10^11^ cp mL^−1^. Multiple measurements yielded high precision of the method with a CV < 5% for the calculated molecular weight of the incorporated transgene. Furthermore, we show that this technique is applicable to different serotypes (AAV5, AAV8 and AAV9) without any method adaptation.

## 2. Results and Discussion

To assess the F/E ratio of an AAV sample, ion exchange chromatography (IEC) combined with UV detection is usually the method of choice. However, the separation of AAV9 capsids using a previously published IEC method coupled to light scattering detectors [28] was not satisfactory, as full and empty species overlapped, probably due to structural changes of the capsid compared to other serotypes. The charge difference between empty and filled AAV vectors is not significant enough to allow a separation of both species, regardless of whether a cation- or anion-exchange approach had been used. Hence, mass photometry provides a simple and fast alternative to quantify both populations in a sample. Additional information on subspecies, like particles containing truncated versions of the genome, is obtained. Compared to IEC, MP requires 5–10 µL of sample, which is one-tenth of the volume used for IEC, omits the need for eluents and is a non-destructive method. In addition, analysis times are reduced to 1–2 min on the mass photometer as opposed to an average 30-min gradient for IEC. A schematic overview of an MP measurement is given in Figure 1. In mass photometry, each event is registered by the instrument, where the intensity of the scattered light is directly proportional to the mass of a capsid [27].

In order to ensure accurate results, clean and dust-free glass slides are required. When reusing microscope coverslips, they were cleaned thoroughly with isopropanol and distilled water before drying them properly with a clean nitrogen stream. Figure 2 depicts the native images of the solid–liquid interface before and after cleaning the glass surface. Furthermore, each sample was diluted prior to measurement to avoid overcrowded microscope images (Figure 2), resulting in lower binding count rates and potential incorrect quantification of AAV vector subpopulations. 

### 2.1. Comparison of % Filled AAV Capsids by Orthogonal Methods

AUC is an orthogonal analytical method to MP for the quantification of empty, partially filled, full and overfull particles. It is based on the different sedimentation velocities of the previously mentioned AAV subpopulations under strong gravitational force due to differences in their size, density, weight and shape [29]. While empty capsids sediment with a sedimentation coefficient (S) between 60 and 64 S, full capsids do so around 100 S. Partially filled AAVs are found in between empty and full populations, while overfilled particles sediment the fastest, hence, have the largest sedimentation coefficient [30]. Consequently, AUC can be applied to confirm the F/E ratio of an AAV sample determined with mass photometry. On the downside, AUC demands large sample volumes and high capsid titers and has low turnaround times and sample throughput [23]. Prompted by this, we implemented mass photometry in our analytics portfolio as a fast and straightforward orthogonal method that provides insight into the homogeneity of an AAV sample. Therefore, two AAV9 samples comprising mostly empty (meC) and mostly filled AAV capsids (mfC), respectively, were mixed at different ratios to obtain fractions of different F/E ratios ranging from 21% to 86% F/E (capsid titers: ~2 × 10^11^ cp mL^−1^). Obtained results are depicted in Figure 3. An excellent linear correlation between MP data (measured % filled) and AUC data (expected % filled) with a coefficient of determination (R^2^) of 0.9949 is observed. This indicates that mass photometry provides a reliable alternative analysis method for the assessment of the F/E ratio. 

### 2.2. Linearity of the Mass Photometer

To assess the sensitivity of the MP, a serial dilution of a sample comprising 1.4% empty, 1.1% partially filled, 86.1% full and 11.4% overfull AAV particles (according to AUC) was conducted, covering a concentration range between 5 × 10^12^ cp mL^−1^ and 4 × 10^10^ cp mL^−1^ (according to ELISA). According to MP manufacturer Refeyn Ltd. (Oxford, UK), 20 µL of a sample with 1 × 10^11^ cp mL^−1^ is the optimal concentration [31]. If the capsid titer is too high, the microscopic image is overcrowded with AAV particles binding and unbinding from the glass slide (Figure 2). The instrument cannot resolve the single events, and a decrease in the binding counts will be observed, jeopardizing the calculations of the molecular weight and the quantification of the proportions of the AAV subpopulations. We have confirmed this concentration-dependent behavior with capsid titers ≥ 2.5 × 10^12^ cp mL^−1^. Figure 4 demonstrates the correlation of the binding count rate with increasing sample concentration. Linearity was observed between 4 × 10^10^ cp mL^−1^ and 8 × 10^11^ cp mL^−1^. Despite not being advised by Refeyn Ltd., we have shown that binding count rates < 1000 (60 s recording time) also were suitable for the determination of the molar mass and F/E ratio with relative standard deviations (RSD) < 6% and errors < 9%. Each dilution was measured in triplicates; however, the binding count rates between the measurements fluctuated quite significantly, capturing the limitations of the instrument. If the counting of the single events were highly reproducible throughout multiple measurements of one sample, it would be possible to determine the capsid titer. Hence, the error bars were omitted in Figure 4. Since mass photometry is based on the detection of single events, the LOD stands and falls with the detection of one single particle, e.g., strongly diluted samples that result in a single measured event. However, we need a significant amount of binding events to determine a F/E ratio. The LOQ was found to be 1.8 × 10^11^ cp mL^−1^ and was calculated using data from Figure 4. Empirically, 8 × 10^10^ cp mL^−1^ was the lowest analyte concentration, which allowed the calculation of the molar mass and the quantification of the subpopulations with a CV < 5%.

### 2.3. Precision of the Mass Photometer

The precision of the MP was tested by measuring an AAV9 sample purified with affinity chromatography (~33% empty, ~6% partially filled, ~52% full and ~8% overfull AAV particles) five times on three consecutive days to additionally check for the impact of repetitive three freeze–thaw (FT) cycles on the AAV integrity. The results show that no significant impact of the FT cycles on the vector integrity was observed regarding the molecular weight of the empty and full AAV species. The average transgene size of 1.45 MDa matches the expected transgene size of 1.40 MDa with an RSD of 2.5% and an error of 3.3% (Table 1). In addition, the measured proportions of the AAV subpopulations are in good agreement with the AUC data. Compared to AUC data, MP underestimates the overfilled AAV populations; however, this trend was consistent for all repetitive measurements (Table 2). A third orthogonal method could be used to confirm the percentage of overfull AAV particles, but this is out of the scope of this study. 

### 2.4. Measurement of Different Serotypes

Since mass photometry is insensitive to the size and shape of detected particles [24,25], it is applicable to various serotypes without adaptation of the instrument configurations. To confirm this, we tested three in-house produced serotypes, AAV5, AAV8 and AAV9. Figure 5 illustrates the variations between the detected masses of the three serotypes. Results were compared with AUC data (Figure 6) with respect to the proportion of the subpopulations (% empty, partially filled, full and overfull). Because of the differences in the genetic cargo of each serotype, the AAV limits had to be set individually. Therefore, samples comprising meC and mfC were selected for each serotype to pre-define the limits of the “empty” and “full” AAV capsid fractions. The mass difference between “empty” and “full” was specified as “partially filled”. Masses beyond “full” were classified as “overfull”. Figure 6 shows that mass photometry agrees well with data obtained by AUC regardless of which serotype (AAV5, AAV8, AAV9) had been used. The percentage of partially filled capsids of AAV8 and AAV9 was accurately captured by the MP, while the percentages of overfilled capsids for AAV5 and AAV9 were slightly underestimated. Nevertheless, obtained results agree well with AUC data highlighting the potential of mass photometry to accurately determine the AAV subpopulations in an AAV sample, regardless of which serotype had been used.

### 2.5. Confirmation of Transgene Size

To further test the performance of the instrument, three in-house produced AAV9 vectors differing in the size of their encapsulated genomic cargo were measured. The molar mass of the nucleic acid of each AAV9 vector was calculated from the 5′-ITR to the 3′-ITR of the respective plasmid using SnapGene software 5.1.5 (GSL Biotech LLC, Chicago, IL, USA) and compared to the calculated transgene sizes determined with MP. The results in Table 3 confirm that MP allows differentiating between AAV vectors with similar transgene sizes. AAV9_a, AAV9_b, and AAV9_c contained transgenes with decreasing sizes ranging from 1400 kDa to 1370 kDa and 890 kDa, respectively (according to SnapGene software). Serotype AAV9_a showed the highest deviation of the calculated transgene size (MP) from the expected one (SnapGene) with an error of 7.4%. In the case of the shortest encapsulated transgene, MP agrees well with the expected data with an error of 0.0%. The capability to distinguish between transgenes of 890, 1370 and 1400 kDa can be attributed to the high resolution of the instrument. As opposed to SEC-MALS, which can also be used for the confirmation of the transgene size, MP does not require any knowledge of the analyte prior to measurement, e.g., the molar extinction coefficient of the nucleic acid. Furthermore, MP provides results within one minute only and does not demand any method of development, omitting long turnaround times as in the case of SEC-MALS.

## 3. Materials and Methods

### 3.1. Sample Preparation

HPLC-grade phosphate-buffered saline (PBS) was purchased from Sigma Aldrich (Saint Louis, MO, USA). AAV samples were produced in-house. Isopropanol, which is used for cleaning the silicon sample gaskets and the glass slides, was obtained from Merck (Darmstadt, Germany). “Empty” AAV9 and Thyroglobulin (TG), which function as calibrants, were purchased from Progen (Heidelberg, Germany) and Sigma Aldrich (Saint Louis, MO, USA), respectively. “Empty” was used in a final concentration of 3.3 × 10^11^ cp mL^−1^, TG at a final concentration of ~100 µg mL^−1^. Samples were prediluted with PBS and further diluted 1:2 in the well of the silicon gasket (Refeyn Ltd., Oxford, UK) installed in the instrument. The final concentrations of the measured samples ranged between 8 × 10^10^ and 2 × 10^11^ cp mL^−1^. No sample preparation was required; however, the samples should not contain major impurities. 

### 3.2. Measurements and Experimental Setup

The AUC measurements were performed at 15,000 rpm and 18 °C on a Beckman AUC Optima instrument (Beckman Coulter, Brea, CA, USA) equipped with an An-50 Ti analytical rotor (Beckman Coulter, Brea, CA, USA) using a total number of 150 scans per sample. The MP measurements were carried out on the SamuxMP instrument (Refeyn Ltd., Oxford, UK). Compared to previous devices such as OneMP and TwoMP, the SamuxMP features a higher resolution tailored especially for the analysis of AAV vectors. The cover glasses and the silicon gaskets containing six sample wells were purchased from Refeyn Ltd. (Oxford, UK). Prior to each measurement a calibration was conducted using the “empty” AAV9 vector (3.74 MDa) and TG (670 kDa). The molar mass of the “empty” AAV9 had been determined by CDMS and was provided by Progen (Heidelberg, Germany). For the generation of the calibration curve, 10 µL of PBS was pipetted into the well of the sample cassette before automatically adjusting the focus and adding 10 µL of AAV9 calibrant to the loaded PBS by mixing it vigorously. The measurement time was set to 60 s resulting in the acquisition of a movie visualizing the binding and unbinding events. The calibrant was measured twice, and the results of both measurements were merged into one calibrant mass. Thyroglobulin was measured under the same conditions as the AAV9 calibrant. 

### 3.3. Data Collection and Processing

The measurements were recorded for 60 s using AcquireMP 2.4.2 (Refeyn Ltd., Oxford, UK) and analyzed with DiscoverMP (v2023 R1.2) (Refeyn Ltd., Oxford, UK). The binding width was set to 40 for all measurements. The ratiometric contrast distribution was fitted by a Gaussian function to obtain the molecular weight of the respective subpopulation. The F/E ratio and linearity were visualized using MATLAB R2020b (MathWorks, Natick, MA, USA).

## 4. Conclusions

In this study, we demonstrated that mass photometry could be used as a fast and simple orthogonal method to the cumbersome and more complex AUC, CDMS and TEM. MP provides a useful non-destructive screening tool operating under native conditions. It offers information on the quantities of empty, partially filled, full and overfull AAV populations and can be used to confirm the molecular weight of the genomic cargo. It does so by using minimal sample volumes and low capsid titers. We managed to calculate the molar mass and quantify AAV subpopulations of samples with capsid titers as low as 8 × 10^10^ cp mL^−1^ with a CV < 5% using just 10 µL total sample volume. In addition, multiple samples can be measured within a short period of time, giving MP an advantage over the more laborious AUC, CDMS and TEM. Good comparability between MP and AUC data was observed regarding the F/E ratio and potential subpopulations. Furthermore, the single event-based light scattering technology allows for the measurement of various serotypes without the adaption of the instrument configurations. The possibility of estimating the capsid titer with high precision based on the number of binding events would favor the mass photometer over ELISA, as the latter is a serotype-dependent analytical technique that requires different antibodies for different serotypes. Additionally, the recent release of a robotic module for the mass photometer paves the way for automated instrument operation and allows a higher sample throughput. The great potential of mass photometry to quickly assess unwanted byproducts (empty, partially filled, and overfilled AAV particles) in an AAV sample alongside the desired full AAV capsids could provide a more user-friendly and less laborious alternative to AUC in the future and allows to make gene therapy products safer for patients. 

## Figures and Tables

**Figure 1 ijms-24-11033-f001:**
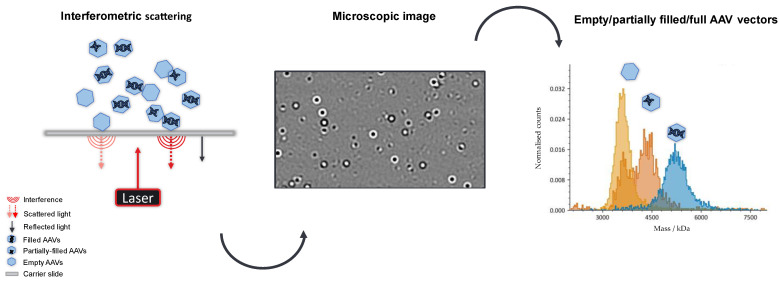
Schematic illustration of a mass photometry measurement. The attachment and detachment of AAVs onto a glass slide result in an interferometric light scattering at the solid–liquid interface. Full and empty particles are visualized in the microscopic image, where white and black circles represent filled and empty AAVs, respectively.

**Figure 2 ijms-24-11033-f002:**
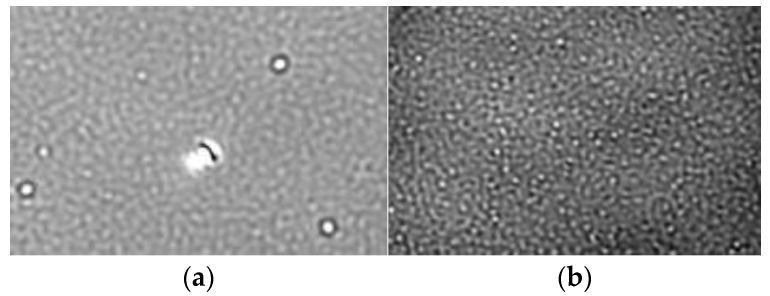
Native microscope images of the solid–liquid interface (**a**) before and (**b**) after cleaning with isopropanol, (**c**) overcrowded with and (**d**) containing an adequate number of AAV particles.

**Figure 3 ijms-24-11033-f003:**
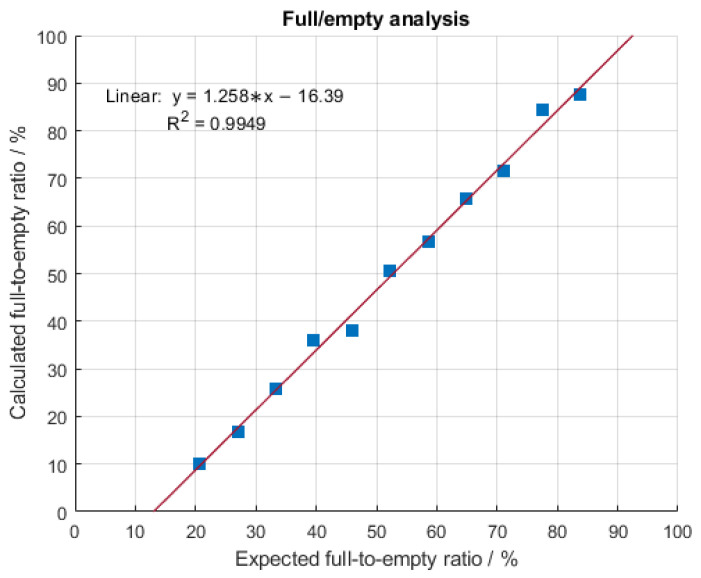
Linear correlation of the measured percentage-filled AAV capsids using mass photometry and expected percentage-filled AAV capsids by AUC.

**Figure 4 ijms-24-11033-f004:**
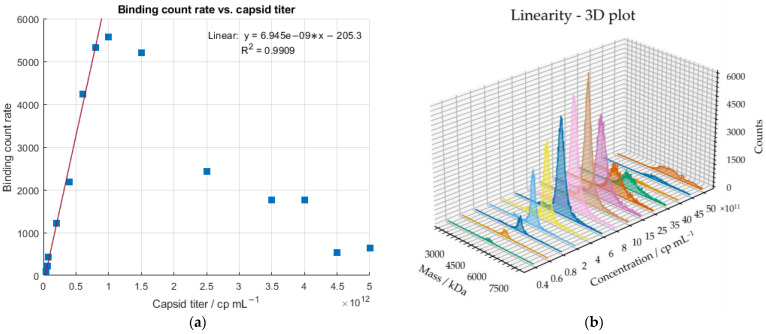
(**a**) Linear correlation between binding count rate and capsid titer (determined with ELISA) for sample concentrations between 4 × 10^10^ cp mL^−1^ and 8 × 10^11^ cp mL^−1^. Sample concentrations ≥ 2.5 × 10^12^ cp mL ^–1^ result in a significant decrease in the binding count rates. (**b**) 3D plot of measured dilutions.

**Figure 5 ijms-24-11033-f005:**
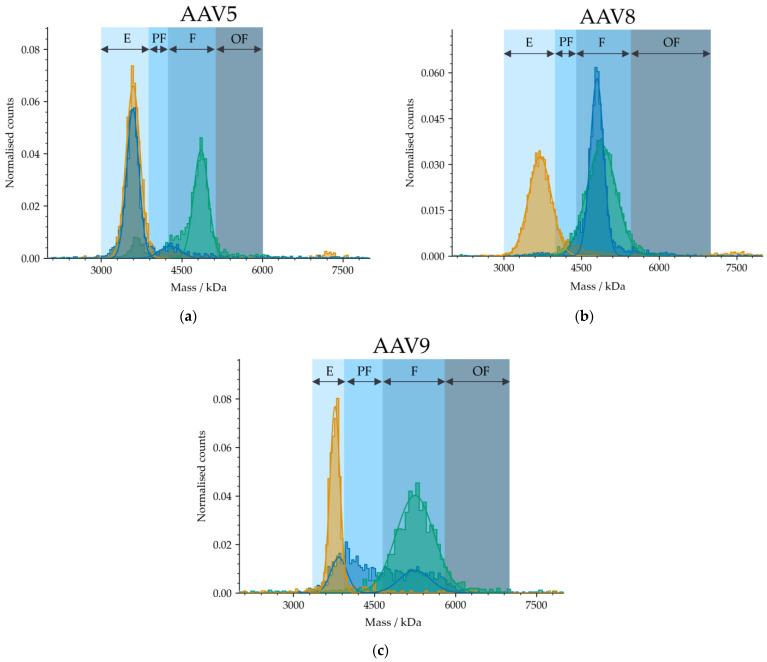
The mass distributions of three AAV serotypes ((**a**) AAV5, (**b**) AAV8, and (**c**) AAV9) comprising different proportions of AAV subpopulations. E, PF, F and OF stand for empty, partially filled, full and overfull, respectively. Samples comprising meC (orange) and mfC (green), respectively, were selected for each serotype to set the AAV limits for the analysis of an AAV sample (blue).

**Figure 6 ijms-24-11033-f006:**
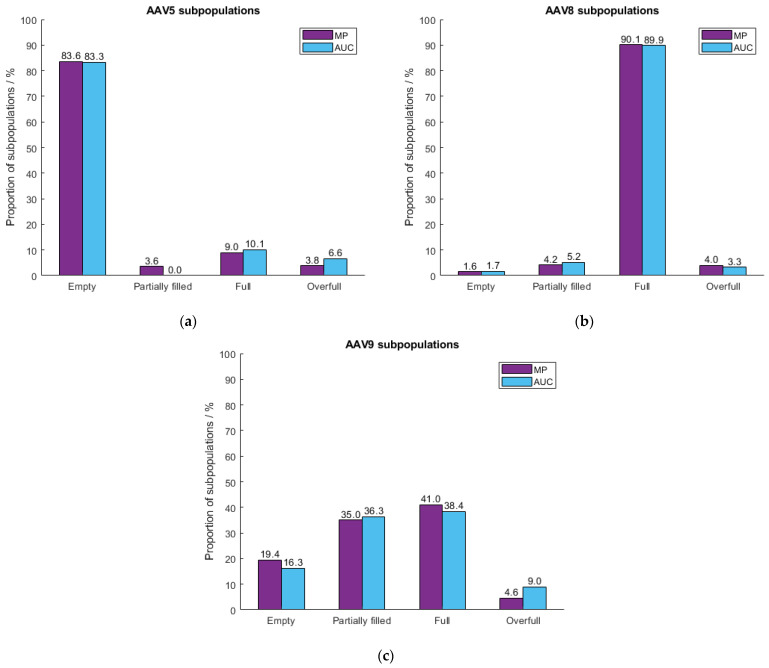
Evaluation of the MP performance using three different in-house produced serotypes (**a**) AAV5, (**b**) AAV8 and (**c**) AAV9. Obtained MP results (purple) were compared to the AUC data (blue).

**Table 1 ijms-24-11033-t001:** Determination of the molar masses of empty and filled subpopulations and confirmation of the transgene size. Data was obtained by five repetitive measurements of an AAV9 sample (~33% empty, ~6% partially filled, ~52% full, and ~8% overfull AAV particles) on three consecutive days (days 1–3).

	Empty AAVs		Full AAVs						
	Average Mass/kDa	RSD/%	Average Mass/kDa	RSD/%	Calculated Transgene Size/kDa	Average Transgene Size/kDa	RSD/%	Expected Transgene Size/kDa	Error/%
Day 1	3583	2.2	4987	2.2	1405	1447	2.5	1400	3.3
Day 2	3746	0.9	5217	0.9	1471				
Day 3	3789	1.3	5253	1.3	1464				

**Table 2 ijms-24-11033-t002:** Determination of the AAV subpopulations of an AAV9 sample comprising ~33% empty, ~6% partially filled, ~52% full, and ~8% overfull particles. The sample was measured five times on three consecutive days (days 1–3). The obtained results were compared to AUC data. E, PF, F and OF stand for empty, partially filled, full and overfull, respectively.

	MP								AUC			
	Average E/%	RSD/%	Average PF/%	RSD/%	Average F/%	RSD/%	Average OF/%	RSD/%	E/%	PF/%	F/%	OF/%
Day 1	39.7	5.4	7.1	8.7	49.7	2.2	3.5	9.8	33.1	6.3	52.3	8.4
Day 2	40.0	6.8	8.8	9.8	47.8	8.6	2.8	5.4				
Day 3	40.4	3.8	8.3	1.8	48.1	3.1	2.3	8.7				

**Table 3 ijms-24-11033-t003:** Comparison of three in-house-produced AAV9 vectors differing in their genomic cargo (a, b, and c).

Serotype	Empty AAVs/kDa	Full AAVs/kDa	Calculated Transgene Size/kDa	Expected Transgene Size/kDa	Error/%
AAV9_a	3750	5253	1503	1400	7.4
AAV9_b	3992	5319	1327	1370	3.1
AAV9_c	3754	4644	890	890	0.0

## Data Availability

Not applicable.

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
