# Peer review of "Quantification of Empty, Partially Filled and Full Adeno-Associated Virus Vectors Using Mass Photometry"

_ijms, 2023, doi:10.3390/ijms241311033_

Round 1

Reviewer 1 Report

This paper describes a case study in the use of mass photometry to determine empty, partial, and full AAV capsid ratios. 

Overall the work is clear and concise and the conclusions are supported by the data. 

Suggested improvements - 

AAV samples typically contain poloxamer 188 to mitigate adsorption losses. In this work it appears that P188 was not present in the samples? Did you perform any studies to determine if the adsorption was in fact reversible to the glass slide in the PBS formulation? 

The capability of other methods could be explained a little more - for example - 

line 62/63 - empty capsids can also be estimated using A260/280 if the extinction coefficient of the protein and the DNA are known, solving for 2 unknowns with 2 measurements - see for instance Porterfield et al. 2010 virology. This method is not mentioned at all in your paper.  Also for example see see Li et al. 2019 Cell and Gene therapy insights. UV has a very rapid turnaround. Limitation of UV being that is cannot assess partially full capsids as it is a bulk technique.

line 67- 'up to' doesn't really make sense as the important point is the lowest concentration that can be measured by AUC. The sensitivity of AUC depends on the wavelength used - less sensitive at 280 nm and more at 230 nm for example. 

The layout of table 3 does not easily allow for a direct comparison - could you present it as two rows for AAV5 for example - the upper being MP and the lower AUC so that the data are together in the same column? 

Since the expected and actual distributions do not match in the linearity plot the results of the method will be offset compared to AUC and therefore might create additional work in terms of generating a correlation for each sample type - can you discuss this in more detail. 

Reviewer 2 Report

Review of Manuscript “Quantification of empty, partially filled and full adeno-associated virus vectors using mass photometry” by Christina Wagner et al. The authors demonstrate that mass spectrophotometry can be used as an orthogonal method to AUC to accurately quantify the proportions of empty, partially filled, full, and overfull particles. Additionally, this method can be used to confirm the molar mass of the packaged genomic material in filled AAV particles. This is very practical and interesting and can attract researchers in this field. There is a need, however, for further revision relating to the aim of the presented studies is reported on. In general, I would recommend to be accepted the paper for publication after minor revision. 

1.       Abstract: Authors may add quantitative or qualitative results of experiments to the Abstract to present the highlights and scientific results of the study.

2.       Introduction: The introduction is very comprehensive, providing valuable background information about AAVs. Page 2, line 84-86. Can the authors please specify how much sample volume and capsid titer is required for this study? Rather than stating the use of minimal sample size and low capsid titers.

3.       Will the authors use this method in the future for studies of biomolecular interactions and kinetic constants such as equilibrium constants (e.g. dissociation Kd), free energy, and rate constants (koff and kon).

4.      Results and Discussion:

(1) Page 3, line 109-110. Compared to IEC, MP uses little sample volume, omits the need for eluents and is a non-destructive method with short analysis times. What is the sample volume? What is the analysis time? Please reshape it.

(2) Figure 3. Linear correlation of measured % filled AAV capsids using mass photometry and expected % filled AAV capsids by AUC.  Experiments were performed several times per point (n=?).

(3) As we all know, the limit of detection (LOD) and limit of quantification (LOQ) are defined as follows: Limit of detection (LOD): The minimum amount of the analyte in the sample that can be distinguished from the signal value of the analytical instrument, but it may not be able to quantify the correct value of the target analyte. Limit of quantification (LOQ): The lowest amount of the analyte in the sample that can be quantitatively measured, and the measurement results have appropriate accuracy and precision. How to calculate the LOD and LOQ in evaluating the sensitivity of MP? Only LOQ is calculated in this paper, while LOD is not estimated.

(4) Page 5, figure 4. Authors should independently plot the linear correlation between binding count rate and capsid titer for sample concentrations between 4 × 10^10 cp mL^– 1 and 8 × 10^11 cp mL^– 1 . In addition, the sample concentration exceeding 8 × 10^11 cp mL^– 1 should belong to the condition of supersaturation or too high concentration leading to decay. This phenomenon has no meaning in the estimation of the limit of detection in biochemical analysis. Please explain why the concentration exceeds 8 × 10^11 cp mL^– 1 .

(5) Is there any sample pretreatment process for the determination of different serotypes, because complex samples should cause non-specific interference. Please explain.

(6) Is section 2.4.1 a typo, or is it section 2.5?

5.       Page 8, line 232. Without the third part, skip directly to the 4. Materials and Methods. Please explain.

6.       Furthermore, the Authors should add something in the conclusion section regarding the scientific implications of their findings and possible future perspectives.

Reviewer 3 Report

ijms-2439181

Quantification of empty, partially filled and full adeno- associated virus vectors using mass photometry

The manuscript by Wagner et al. described the use of mass photometry as an orthogonal method to analytical ultracentrifugation to quantify the proportions of empty, partially filled, full, and over-full particles in Adeno-associated viruses (AAV) samples. Overall, the data were well presented and sufficient for the conclusion. However, there are some issues to consider as follows.

1. Materials and Methods section: The authors should describe the standard analytical ultracentrifugation used for obtaining data in Tables 2 and 3.

2. The manuscript lacks discussion. Most of the data were presented without intensive discussion. The authors should improve it and cite relevant references.

3. The authors should expand the discussion of the data in Tables 3 and 4.

4. There was no section 3. Please renumber all sections.

5. Was (a part of) Figure 1 adapted from reference #27? Please clarify. If yes, please include a statement of copyright and permission to reprint.

Round 2

Reviewer 3 Report

The manuscript was revised but still had some minor issues to improve, as follows.

1. Materials and Methods section: The authors should describe the standard analytical ultracentrifugation (speed, time, temperature, and instrumentation).

2. References #5, 6, 7, 29, and 30 in the Reference list should follow the journal guidelines.

2. 

Author Response

We have updated the manuscript:

  1. Materials and Methods section: The authors should describe the standard analytical ultracentrifugation (speed, time, temperature, and instrumentation).
    The AUC measurements were performed at 15,000 rpm and 18 °C on a Beckman AUC Optima instrument (Beckman Coulter, Brea, CA, USA) equipped with an An-50 Ti analytical rotor (Beckman Coulter) using a total number of 150 scans per sample.
  2. References #5, 6, 7, 29, and 30 (31?) in the Reference list should follow the journal guidelines.
    References were updated but will also be adapted to the guidelines of the journal when undergoing final format changes.